# The Edible Brown Seaweed *Sargassum horneri* (Turner) C. Agardh Ameliorates High-Fat Diet-Induced Obesity, Diabetes, and Hepatic Steatosis in Mice

**DOI:** 10.3390/nu13020551

**Published:** 2021-02-08

**Authors:** Shigeru Murakami, Chihiro Hirazawa, Takuma Ohya, Rina Yoshikawa, Toshiki Mizutani, Ning Ma, Mitsuru Moriyama, Takashi Ito, Chiaki Matsuzaki

**Affiliations:** 1Department of Bioscience and Biotechnology, Fukui Prefectural University, Fukui 9101195, Japan; s1973013@g.fpu.ac.jp (C.H.); s1621009@g.fpu.ac.jp (T.O.); s1621044@g.fpu.ac.jp (R.Y.); s1621035@g.fpu.ac.jp (T.M.); tito@fpu.ac.jp (T.I.); 2Division of Health Science, Graduate School of Health Science, Suzuka University, Mie 5100293, Japan; maning@suzuka-u.ac.jp; 3Fukui Food Processing Research Institute, Fukui 9100343, Japan; m-moriyama-m1@pref.fukui.1g.jp; 4Research Institute for Bioresources and Biotechnology, Ishikawa Prefectural University, Ishikawa 9218836, Japan; chiaki@ishikawa-pu.ac.jp

**Keywords:** obesity, diabetes, hepatic steatosis, high-fat diet, lipase inhibition, seaweed, *Sargassum horneri* (Turner) C. Agardh

## Abstract

*Sargassum horneri* (Turner) C. Agardh (*S. horneri*) is edible brown seaweed that grows along the coast of East Asia and has been traditionally used as a folk medicine and a local food. In this study, we evaluated the effects of *S. horneri* on the development of obesity and related metabolic disorders in C57BL/6J mice fed a high-fat diet. *S. horneri* was freeze-dried, fine-powdered, and mixed with a high-fat diet at a weight ratio of 2% or 6%. Feeding a high-fat diet to mice for 13 weeks induced obesity, diabetes, hepatic steatosis, and hypercholesterolemia. Supplementation of mice with *S. horneri* suppressed high-fat diet-induced body weight gain and the accumulation of fat in adipose tissue and liver, and the elevation of the serum glucose level. In addition, *S. horneri* improved insulin resistance. An analysis of the feces showed that *S. horneri* stimulated the fecal excretion of triglyceride, as well as increased the fecal polysaccharide content. Furthermore, extracts of *S. horneri* inhibited the activity of pancreatic lipase in vitro. These results showed that *S. horneri* can ameliorate diet-induced metabolic diseases, and the effect may be partly associated with the suppression of intestinal fat absorption.

## 1. Introduction

Seaweed has few calories and contains a wide variety of nutritional components including protein, polysaccharides, unsaturated fatty acids, minerals, vitamins, and amino acids [1,2,3]. Some of these nutrients are found at higher levels in seaweed than in terrestrial plant-derived foods. In addition, seaweed is rich in bioactive compounds, such as polyphenols and carotenoids [3]. Recent epidemiological evidence indicates that a seaweed intake is associated with a reduced incidence of cardiovascular disease mortality and increased life expectancy [4,5]. Seaweed has been regularly consumed in the daily diet since ancient times in Japan, Korea, and China, which may contribute to the longevity noted in these countries [6].

Obesity is a chronic metabolic disorder characterized by abnormal or excessive fat accumulation in the body. The prevalence of obesity has increased worldwide, reaching pandemic levels [7]. Obesity has been linked to a number of serious diseases, such as type 2 diabetes, cardiovascular disease, fatty liver disease, sleep apnea, and some cancers [8,9]. Various kinds of compounds, including polyphenols, polyunsaturated fatty acids, and dietary fiber, from fruits, vegetables, grains, seaweed, and medical plants have been reported to exhibit an anti-obesity effect in experimental animal models [10,11,12,13]. Although these compounds are expected to help attenuate the development of obesity, human data are scarce, in contrast to the large number of animal experiments that have been performed.

*Sargassum horneri* (Turner) C.Agardh (*S. horneri*), also known as Akamoku in Japan, is a brown seaweed that grows on the coast of East Asia [14]. *S. horneri* has been used as a food source and traditional medicine to treat several disorders for centuries in Japan, Korea, and China [15]. In Japan, *S. horneri* has been used as a local foodstuff in the Tohoku region since ancient times. Recently, *S. horneri* has been attracting increasing attention, as several studies using animal models have shown that higher levels of some active constituents exert beneficial effects on health promotion and disease prevention [16,17]. *S. horneri* contains high concentrations of polysaccharides, such as fucoidan and alginate. Fucoidan, a sulfated fucose-containing polysaccharide, has been shown to have a wide variety of biological activities, including anticancer, anticoagulant, immune-regulatory, anti-inflammatory, antiviral, anti-obesity, and antidiabetic effects in animal and in vitro studies [18,19]. The algal polysaccharide alginate has been isolated from the cell walls of brown seaweed and widely used in the food industry as a stabilizer or emulsifying agent [20]. Alginate acts as a dietary fiber and prevents the progression of cardiovascular and gastrointestinal diseases [21,22]. In human studies, alginate has been reported to suppress hunger and reduce the percentage of body fat [23,24]. Fucoxanthin, a marine carotenoid mainly found in brown seaweed, being especially rich in *S. horneri*, has been shown to exert multiple biological effects, including antioxidant, anticancer, anti-inflammatory, anti-angiogenic, anti-obesity and antidiabetic activities [25]. Thus, *S. horneri* has high levels of active components, and the health benefits of each component have been demonstrated in experimental animals and humans. However, few data on the preventive effects of whole *S. horneri* on metabolic diseases are available. It is important to understand the health-promoting effect of whole *S. horneri* as a traditional foodstuff. In the present study, we evaluated whether or not *S. horneri* could ameliorate the development of obesity and its related diseases in mice fed a high-fat diet.

## 2. Materials and Methods

### 2.1. Preparation of Powder and Extracts of S. horneri

The *S. horneri* sample was harvested on the coast of Fukui Prefecture in April 2019 and washed with water. It was freeze-dried, reduced to a fine powder using a food mixer, and mixed into a high-fat diet (HF). For enzyme inhibition experiments, powdered *S. horneri* was extracted with ethanol or water. Ethanol extract was prepared by homogenizing the sample powder in 70% ethanol and left at room temperature for two days. The sample was centrifuged at 3000× *g* for 15 min. The supernatant was then evaporated to dryness. Water extract was prepared by homogenizing the sample powder with water and left in boiling water for 30 min. The sample was centrifuged at 3000× *g* for 15 min and lyophilized.

### 2.2. Animal Experiments and Dietary Treatment

Six-week-old male C57BL/6J mice were purchased from CLEA Japan Inc. (Tokyo, Japan) and housed in a controlled atmosphere (22 ± 1 °C at 50% relative humidity) with a 12 h light/dark cycle. After 1 week of acclimation, the animals were randomly divided into four groups, as follows: (1) normal diet (Normal) group (*n* = 13), (2) high-fat diet (HF) group (*n* = 12), (3) group with a HF diet supplemented with 2% *S. horneri* (HF + *S. horneri* low-dose (ShL) group) (*n* = 12), and (4) a group with a HF diet supplemented with 6% *S. horneri* (HF + *S. horneri* high-dose (ShH) group) (*n* = 12). The compositions of the experimental diets were adjusted by considering the nutritional components of *S. horneri* (Table 1). The normal diet provided 354 kcal/100 g of energy (14.4% calories from protein, 11.1% calories from fat, and 74.4% calories from carbohydrate), while the HF provided 493 kcal/100 g of energy (17.9% calories from protein, 60.7% calories from fat, and 21.4% calories from carbohydrate). All experimental diets were based on the AIN-76 diet (Oriental Yeast Co. Ltd., Tokyo, Japan). Animals were allowed free access to diets and drinking water. The body weight and food intake were monitored every other day. After 13 weeks of feeding, the mice were deprived food overnight, and blood samples were withdrawn from the ophthalmic vein under a mixed anesthetic agent (0.3 mg/kg of medetomidine, 4.0 mg/kg of midazolam, and 5.0 mg/kg of butorphanol; Fujifilm Wako Pure Chemical Co., Osaka, Japan). Liver tissue and epididymal, peritoneal, and mesenteric white adipose tissues were removed, weighed, and stored at −80 °C. Some of the mice (*n* = 3−4) were used for histological examinations. All experimental protocols were approved by the Institutional Animal Care and Use Committee of Fukui Prefectural University (approval No. 19–14).

### 2.3. Serum Biochemical Analyses

Serum was obtained by centrifugation at 1500× *g* for 15 min at 4 °C. The serum levels of total cholesterol, high-density lipoprotein (HDL)-cholesterol, triglyceride, alanine aminotransferase (ALT), aspartate aminotransferase (AST), alkaline phosphatase (ALP), and leucine aminopeptidase (LAP) were analyzed using a Hitachi 7060 Automatic Analyzer (Hitachi, Tokyo, Japan) with commercial kits (Fujifilm Wako Pure Chemical Co., Osaka, Japan). The non-HDL cholesterol levels were calculated by subtracting the HDL-cholesterol from the total cholesterol. Serum insulin (Morinaga Institute of Biological Science, Yokohama, Japan), adiponectin (Otsuka Pharmaceutical Co. Ltd., Tokyo, Japan), and tumor necrosis factor-α (TNF-α; Fujifilm Wako Pure Chemical Co., Osaka, Japan) levels were also determined using a commercial ELISA kit.

### 2.4. Liver Lipid Analyses

Lipids were extracted from the liver according to the method described previously [26]. In brief, the frozen liver tissues (50 mg) were homogenized (20%, w/v) in isopropanol. The homogenate was kept at room temperature for 2 days and then centrifuged at 1000× *g* for 10 min. Aliquots of the supernatant were analyzed for triglyceride content using a commercial kit (Fujifilm Wako Pure Chemical Co., Osaka, Japan).

### 2.5. Histological Analyses

The mice were anesthetized with an intraperitoneal injection of mixed anesthetic agent (0.3 mg/kg of medetomidine, 4.0 mg/kg of midazolam, and 5.0 mg/kg of butorphanol; Fujifilm Wako Pure Chemical Co., Osaka, Japan), and perfused transcardially with a fixative containing 4% paraformaldehyde and 1.5% glutaraldehyde in phosphate-buffered saline (PBS). After the perfusion, the liver and white adipose tissue were removed and allowed to stand in the same fixative for one day. The tissues were rinsed several times with PBS and embedded in paraffin. Sections of tissues were cut into 5-μm-thick sections, mounted on slides, and stained with hematoxylin eosin (HE).

### 2.6. Glucose Tolerance Test

A glucose tolerance test was performed one week before the end of experiment. The mice fasted overnight and were intraperitoneally injected with glucose (2 g/kg body weight). The blood samples were collected from the tail veins of the mice, and glucose levels were measured at 0, 30, 60, 90, and 120 min after injection using a blood glucometer Nipro Stat Strip (Nipro, Osaka, Japan).

### 2.7. Fecal Analyses

During the fecal collection, mice were separated, and fecal samples for a 24-h period were collected from each mouse and weighed. These samples were ground into a powder in a mortar, and 50 mg of feces was extracted with 300 μL of distilled water. After centrifugation (16,000× *g*, 30 min, 4 °C), ethanol was added to the supernatant (final concentration of 85%), and polysaccharides were obtained as the precipitate. The resulting residue was washed with 85% ethanol and dried. The residue was then resuspended in distilled water and centrifuged (16,000× *g*, 10 min, 4 °C), and polysaccharide content in the supernatant was measured using a phenol-sulfuric acid method described elsewhere with galactose as the standard [27]. For the measurement of triglycerides, lipids were extracted by adding isopropanol (10 times the weight) to the fecal powder. The sample was then dried and dissolved in isopropanol. The concentration of triglyceride was measured using a commercial kit (Fujifilm Wako Pure Chemical Co., Osaka, Japan).

### 2.8. Lipase Assay

The inhibitory activity of *S. horneri* extracts on lipase was determined as previously described [28] with some modification. In brief, lipase (type II, from porcine pancreas, 400 units/mg protein; Sigma-Aldrich Corp., Saint Louis, MO, USA) was dissolved in distilled water at 5 mg/mL and then centrifuged (1000× *g*, 5 min), and the supernatant was used as the enzyme source. Next, 4-Nitrophenyl butyrate (4-NPB; Sigma-Aldrich Corp., Saint Louis, MO, USA) was dissolved in dimethylsulfoxide. The reaction mixture contained 100 μL of enzyme solution and 100 μL of *S. horneri* extract in 4 mL of 20 mM Tris-HCl buffer pH 8.5. The mixture was pre-incubated at 37 °C for 10 min. The reaction was started by the addition of 100 μL of 5 mM 4-NPB solution and then incubated for 30 min at 37 °C. The absorbance was measured at 400 nm.

### 2.9. Statistical Analysis

Data are expressed as mean ± SEM. Statistical analysis was performed using one-way ANOVA followed by Tukey’s multiple range test. All of the results were considered statistically significant at p<0.05.

## 3. Results

### 3.1. Effects of S. horneri on Food Intake and Body Weight

The addition of *S. horneri* to an HF diet did not affect the amount of food consumption. The average daily food intake of each group throughout the experimental period was 3.1 g (Normal), 2.4 g (HF), 2.6 g (HF + ShL), and 2.4 g (HF + ShH). The energy intake of mice on the normal diet was not significantly different from those with the HF (11.9 vs. 12.6 kcal/mouse/day). Feeding HF to mice for 13 weeks induced marked weight gain. The HF mice showed a significantly higher body weight gain than the Normal mice (Figure 1b). The body weights began to differ significantly between the HF and HF + ShH mice after 4 weeks of treatment (Figure 1a). The body weights between the HF and HF + ShL mice differed significantly after 10 weeks of treatment. In the last week of the experiment, the body weight for both doses of *S. horneri* was significantly lower than that in the HF mice (Figure 1a).

### 3.2. Effects of S. horneri on the Mass and Morphology of White Adipose Tissue

Consistent with the increase in body weight, feeding an HF to mice significantly increased the weight of white adipose tissue, including epididymal, retroperitoneal, and mesenteric adipose tissues, compared to those in the Normal group (Figure 2a). The total fat weight of the HF group was also significantly higher than that of the Normal group. Thirteen-week-treatment of mice with *S. horneri* suppressed the HF-induced increase in fat weight in all white adipose tissues examined, including epididymal, retroperitoneal, and mesenteric adipose tissues. A morphological analysis by hematoxylin and eosin (HE) staining showed that adipocytes were enlarged in the HF group compared to those in the Normal group (Figure 2b). In contrast, the adipocyte size was smaller in the *S. horneri*-treated mice than in the HF group.

### 3.3. Effects of S. horneri on Serum Levels of Glucose, Insulin and Adipokines, and Insulin Resistance

Serum glucose and insulin levels were significantly increased by HF diet (Figure 3a,b). Supplementation with *S. horneri* suppressed the increased levels of glucose and insulin in a dose-dependent manner (Figure 3a,b). The effect of *S. horneri* on insulin resistance was assessed using an intraperitoneal glucose tolerance test. The peak of blood glucose was lower, and the glycemic response was improved in *S. horneri*-treated mice, compared to mice in the HF group (Figure 3c). The area under the curve (AUC) for glucose was also significantly lower in *S. horneri*-treated mice compared to mice in the HF group (Figure 3d). Serum level of anti-inflammatory adiponectin was significantly reduced by feeding of HF diet (Figure 4a), while inflammatory cytokine TNF-α level was markedly elevated by HF diet (Figure 4b). Supplementation with *S. horneri* normalized these changes in a dose-dependent manner.

### 3.4. Effects of S. horneri on Serum Lipid Levels

The serum levels of total cholesterol and non-HDL cholesterol were significantly increased in the HF group compared to those in the Normal group (Table 2). By contrast, the levels of HDL cholesterol and triglyceride were unchanged by feeding an HF. Although low-dose *S. horneri* had little effect on the total cholesterol and non-HDL levels in the serum, high-dose *S. horneri* significantly decreased these parameters.

### 3.5. Effects of S. horneri on Hepatic Steatosis

Feeding an HF to mice increased the liver weight, accompanied by the marked elevation of the liver triglyceride content (Figure 5a,b). Supplementation with *S. horneri* significantly suppressed the increase in the liver weight. It also suppressed the liver triglyceride accumulation in a dose-dependent manner. Gross morphology showed that the liver of the HF group was larger and exhibited a paler color than the Normal group (Figure 5c). A histological examination revealed that feeding an HF for 13 weeks caused hepatic steatosis, as evidenced by vacuoles, lipid droplets, and hepatocyte swelling (Figure 5c). Treatment of HF mice with *S. horneri* attenuated these pathological changes. In particular, HF + ShH normalized HF-induced hepatic steatosis. Consistent with histological observations, the ingestion of an HF increased the serum parameters of the liver function, including ALT, AST, ALP, and LAP (Table 3). HF + ShH decreased these elevated parameters to the normal level.

### 3.6. Effects of S. horneri on the Fecal Content of Triglycerides and Polysaccharides

An HF increased the triglyceride content in the feces more than three-fold compared to the normal mice (Figure 6a). High-dose *S. horneri* increased the fecal triglyceride levels significantly further. In addition, treatment with *S. horneri* significantly increased the fecal content of polysaccharides (Figure 6b).

### 3.7. Effects of S. horneri on the Pancreatic Lipase Activity

Two different extracts were prepared from *S. horneri*, and their inhibitory activity on pancreatic lipase was tested in vitro. Both water extract and ethanol extract dose-dependently inhibited the activity of lipase, with IC_50_ values of 3.7 mg/mL and 2.3 mg/mL, respectively (Figure 7).

## 4. Discussion

The present study showed that dietary supplementation with *S. horneri* was able to ameliorate the development of obesity and related metabolic disorders, including diabetes, hepatic steatosis, and hypercholesterolemia, in mice fed a high-fat diet (HF). *S. horneri* is characterized by a higher content of bioactive polysaccharides and fucoxanthin than other popular edible brown seaweeds, including Wakame seaweed (*Undaria pinnatifida*) and Japanese tangle (*Saccarina japonica*). We confirmed that the *S. horneri* used in the present study contained approximately 35−52% (wt) alginate and 5−11% (wt) fucoidan as polysaccharides. The anti-obesity effects of these components have been previously reported. The effect of the sulfated polysaccharide fucoidan on obesity was examined in mice fed an HF [29]. This animal experiment revealed that fucoidan reduces body weight gain, epididymal fat mass, plasma triglyceride, and liver steatosis, which are accompanied by the down-regulation of the mRNA expression of PPARγ, adipose-specific fatty acid-binding protein, and acetyl CoA carboxylase in adipose tissue. In 3T3-L1 adipocytes, fucoidan ameliorates the lipid accumulation by increasing the expression of hormone-sensitive lipase, a key enzyme involved in lipolysis, together with suppression of inflammation and reactive oxygen species [30]. In vitro and animal studies have suggested that the anti-obesity effects of fucoidan are associated with antioxidative and anti-inflammatory action and enhanced lipolysis. A randomized, double-blind, placebo-controlled study showed the beneficial effects of fucoidan on insulin secretion and serum cholesterol levels in overweight or obese adults [31]. Another major polysaccharide in *S. horneri*, alginate, has also been shown to have anti-obesity effects. Alginate inhibits the activity of pancreatic lipase, leading to decreased breakdown and retarded absorption of triacylglycerol [32]. Furthermore, alginate is a gelling polysaccharide that increases satiety and reduces energy intake [33]. Several studies, including human experiments have suggested that these effects of alginate may be related to delayed gastric clearance and the stimulation of stretch receptors leading to attenuated fat absorption.

The anti-obesity effect of *S. horneri* was demonstrated by a decrease in the body-weight gain and visceral-fat weight in the present HF-induced obese mice. It is well known that white adipose tissue is an active endocrine organ that expresses and secretes various adipokines such as adiponectin, leptin, tumor-necrosis factor-α (TNF-α), interleukin-6 (IL-6), and monocyte chemoattractant protein-1 (MCP-1) [34,35]. These factors play significant roles in the regulation of energy, glucose, and lipid metabolism. The hypertrophy of adipocytes during excessive visceral-fat accumulation causes the alteration of the adipokine-secretion pattern and induces an inflammatory condition that contributes to the onset of obesity-related comorbidities. Adipose-tissue macrophages are also involved in obesity-related inflammation and systemic insulin resistance [31,32]. Thus, it is suggested that an increase in visceral adipose tissue induced insulin resistance, hepatic steatosis, and dyslipidemia in mice fed an HF.

Adiponectin is the most abundant adipokine that exerts insulin-sensitizing actions in obesity-related metabolic disorders. The circulating adiponectin level is inversely related to metabolic dysregulation, the inflammatory process, and oxidative stress and is lower in obese subjects and animals with insulin resistance than healthy subjects and animals [36]. Although ingestion of an HF for 13 weeks reduced the serum adiponectin level, treatment with *S. horneri* suppressed this decrease in a dose-dependent manner. Consistent with previous findings, the present study indicates a negative correlation between the serum adiponectin level and obesity-related metabolic disorders, including diabetes, hepatic steatosis, and dyslipidemia in mice. TNF-α is an important pro-inflammatory cytokine that is critically involved in the development of insulin resistance and pathogenesis of type 2 diabetes [34,35]. Our previous study using the same HF-induced obese mice indicated that elevation of the serum TNF-α level was correlated with the increased mRNA expression of TNF-α in white adipose tissue [37]. Although the serum level of TNF-α was markedly increased by the ingestion of an HF, treatment with *S. horneri* suppressed this increase in serum TNF-α, suggesting that *S. horneri* may improve the inflammatory condition in adipocytes. In fact, ethanol extract from *S. horneri* has been shown to exert an anti-inflammatory effect. *S. horneri* extract exerts anti-inflammatory actions in RAW 264.7 macrophage cells through the inhibition of ERK, *p*-p38, NF-κB, and pro-inflammatory gene expression [38]. In addition, the anti-inflammatory effects of major components of *S. horneri* (e.g., fucoidan [39], alginate [40], and fucoxanthin [41]), have also been reported. Thus, supplementation with *S. horneri* may alleviate inflammation in white adipose tissue, which is partly associated with its anti-obesity effect. Furthermore, it is also well documented that adipose-tissue-derived adipokines play a central role in the development of hepatic steatosis by causing liver inflammation [42].

The present in vitro study showed that both water extract and ethanol extract of *S. horneri* inhibited the activity of pancreatic lipase. Although we did not perform a component analysis of *S. horneri,* water extract is presumed to mainly contain polysaccharides, such as alginate and fucoidan, while 70% ethanol extract is expected to be rich in polyphenols. Hundreds of extracts isolated from a wide variety of plants, seaweed, and bacteria have been reported to inhibit the activity of pancreatic lipase [43]. The active compounds include polyphenols, triterpenes, saponins, and polysaccharides. Some brown seaweed extracts, including *Ascophyllum nodosum*, *Fucus vesuculosus*, and *Pelvetia canaliculata*, have been shown to inhibit lipase activity [44]. Extracts of these seaweeds contain polysaccharides, such as alginate, fucoidan, and laminarin, as well as low-molecular-weight active compounds, such as polyphenols. These polysaccharides and polyphenols may be mainly involved in the inhibition of lipase activity. Alginate has been shown to inhibit the activity of pancreatic lipase and increase fat excretion in rats [45] and human [46]. Several possible mechanisms concerning the inhibitory effect of alginate on lipase have been reported. Alginate may interact with both the substrate and the enzyme through electrostatic interactions, as negatively charged alginate can associate with positively charged proteins [47]. Furthermore, alginate has been shown to decrease the diffusion of lipid in the porcine intestinal mucus layer, leading to reduced lipid absorption [48]. Thus, certain active components of *S. horneri*, including polysaccharides and polyphenols, are suggested to inhibit pancreatic lipase.

Pancreatic lipase is responsible for the absorption of dietary fat, hydrolyzing triacylglycerols to monoacylglycerols, and fatty acids [49]. Since excess dietary fat is the major source of undesirable calories, the inhibition of this enzyme is a possible mechanism by which *S. horneri* reduces fat absorption and ameliorates obesity. Orlistat, the only authorized anti-obesity drug, acts by inhibiting digestive lipase, reducing hydrolysis of ingested fat, and thereby increasing fecal-fat excretion [50]. Our results indicate that *S. horneri* inhibits pancreatic lipase and thereby suppresses the hydrolysis of fat, leading to reduced fat absorption. A reduction in fat absorption results in not only the amelioration of obesity but the improvement of insulin resistance, hepatic steatosis, and dyslipidemia. Analyses of feces revealed that the fecal triglyceride content was markedly elevated by high-dose *S. horneri*, accompanied by an increase in the fecal polysaccharide content. Elevation of the fecal polysaccharide levels suggests that alginate and fucoidan play an important role in the inhibition of intestinal fat absorption and obesity development by *S. horneri.*

Fucoxanthin is a characteristic marine carotenoid present in brown seaweed. It should be noted that the content of fucoxanthin in *S. horneri* is higher than in other brown seaweed [51]. Fucoxanthin exhibits various benefits including anti-obesity and antidiabetic activities [52]. Sixteen-week, double-blind, randomized, placebo-controlled studies have demonstrated the effectiveness of fucoxanthin in the treatment of obese humans [53]. Fucoxanthin and its metabolite fucoxanthinol have also been reported to inhibit the activity of lipase [54]. The half-maximal inhibitory concentration for enzymatic activity (IC_50_) was around 700 nmol/L, which is 100-fold stronger than that of orlistat. The inhibition of triglyceride absorption in vivo was demonstrated in conscious rats. These previous findings support the notion that active components characteristic of *S. horneri*, including alginate, fucoidan, and fucoxanthin, may be involved in the inhibition of pancreatic lipase, which results in the stimulation of fecal-fat excretion and suppression of fat absorption.

In addition to the effect of fucoxanthin on lipase activity, several studies have shown that the anti-obesity effect of fucoxanthin is associated with fatty-acid oxidation and heat production by inducing uncoupling protein 1 (UCP-1) in white adipose tissue [55]. Nutrigenomic studies have shown that fucoxanthin induces UCP1 in the mitochondria of abdominal white adipose tissue, leading to the oxidation of fatty acids and heat production. Fucoxanthin improves insulin resistance and decreases blood glucose levels through the regulation of cytokine secretion from white adipose tissue [56].

In the present study, supplementation of mice with *S. horneri* decreased the HF-induced increase in serum non-HDL cholesterol levels. Possible mechanisms include a reduced intestinal reabsorption and an enhanced fecal excretion of bile acids. Dietary fiber is known to interact with bile acids and interfere with their reabsorption in the intestine. As a result, the fecal excretion of bile acids increases. Decreased intestinal circulation of bile acids up-regulates bile acid synthesis from cholesterol, which results in a decreased serum cholesterol level.

## 5. Conclusions

In conclusion, the present results show that whole *S. horneri* suppressed the development of HF-induced obesity and related metabolic disorders in mice. The anti-obesity effect of *S. horneri* is associated with the inhibition of pancreatic lipase, which leads to the suppression of intestinal lipid absorption and their subsequent accumulation in adipose tissue and liver. In addition, the anti-inflammatory action may be partly involved in the improvement of obesity and diabetes. The major components of *S. horneri* fucoidan, alginate, and fucoxanthin seem to be responsible for its beneficial effects. Thus, the traditionally eaten brown seaweed *S. horneri* may be useful as a foodstuff for reducing rates of obesity and diabetes. This study involved experiments using mice, so the results cannot be directly applied to humans. Further studies are needed to demonstrate the effectiveness of *S. horneri* in humans.

## Figures and Tables

**Figure 1 nutrients-13-00551-f001:**
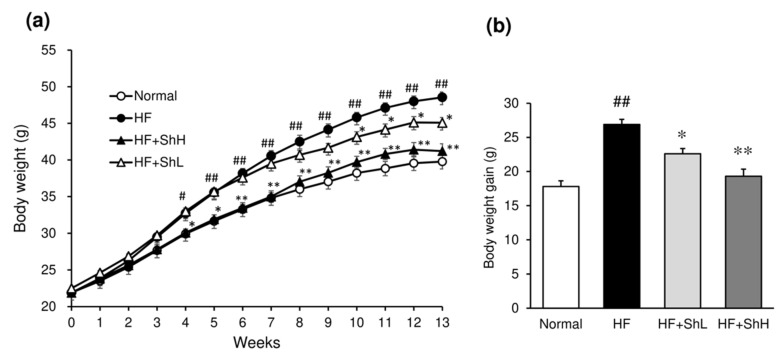
Effects of *S. horneri* on the body weight in C57BL/6J mice fed a high-fat diet for 13 weeks. (**a**) Weekly changes in the body weight; (**b**) body weight gain. Normal, normal diet; high-fat (HF), high-fat diet; HF + ShL, high-fat diet mixed with 2% *S horneri*; HF + ShH, high-fat diet mixed with 6% *S. horneri*. Each value represents the mean ± SEM (*n* = 12−13). Significant difference: * *p* < 0.05, ** *p* < 0.01 vs. HF group, # *p* < 0.05, ## *p* < 0.01 vs. Normal group.

**Figure 2 nutrients-13-00551-f002:**
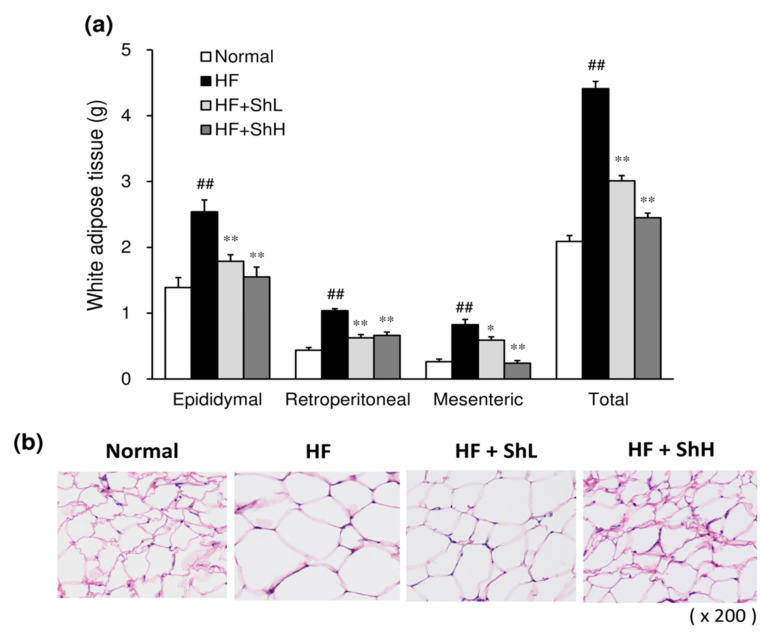
Effects of *S. horneri* on the mass and morphology of white adipose tissue in C57BL/6J mice fed a high-fat diet for 13 weeks. After the mice were sacrificed, masses of epididymal, retroperitoneal, and mesenteric white adipose tissue were determined (**a**). Adipose tissue was fixed, and the section of epididymal adipose tissue was stained with hematoxylin and eosin (HE) (**b**). Normal, normal diet; HF, high-fat diet; HF + ShL, high-fat diet mixed with 2% *S. horneri*; and HF + ShH, high-fat diet mixed with 6% *S. horneri*. Each value represents the mean ± SEM (*n* = 8). Significant difference: * *p* < 0.05, ** *p* < 0.01 vs. HF group, ## *p* < 0.01 vs. Normal group.

**Figure 3 nutrients-13-00551-f003:**
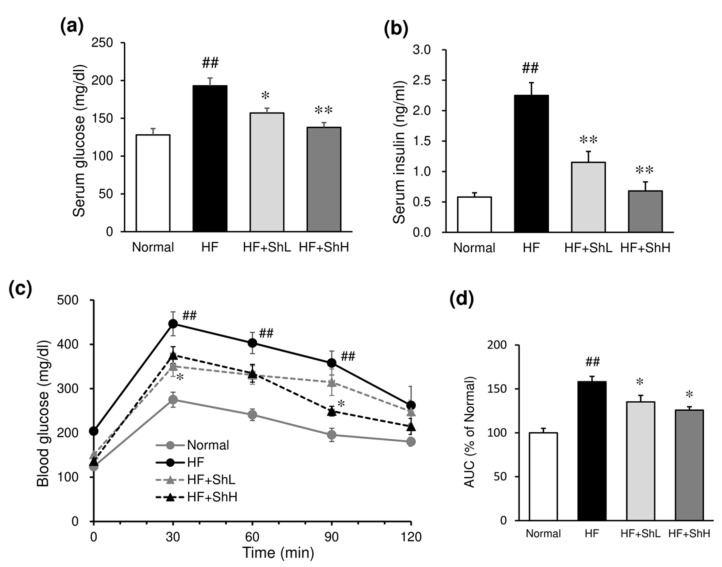
Effects of *S. horneri* on serum glucose and insulin levels and on the insulin resistance in C57BL/6J mice fed a high-fat diet. Serum levels of glucose (**a**) and insulin (**b**) were determined 13 weeks after high-fat diet ingestion. Insulin resistance (**c**) was evaluated in glucose tolerance test on the 12th week after high-fat diet ingestion, and the area under the curve (AUC) was calculated (**d**). Normal, normal diet; HF, high-fat diet; HF + ShL: high-fat diet mixed with 2% *S horneri*; HF + ShH: high-fat diet mixed with 6% *S horneri*. Each value represents the mean ± SEM (*n* = 6−8). Significant difference: * *p* < 0.05, ** *p* < 0.01 vs. HF group, ## *p* < 0.01 vs. Normal group.

**Figure 4 nutrients-13-00551-f004:**
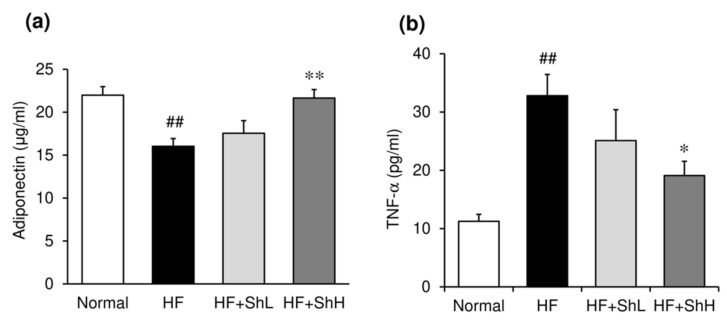
Effects of *S. horneri* on serum adipokine levels in C57BL/6J mice fed a high-fat diet for 13 weeks. Serum levels of adiponectin (**a**) and TNF-α (**b**) were determined using a commercial ELISA kit. Normal, normal diet; HF, high-fat diet; HF + ShL, high-fat diet mixed with 2% *S horneri*; HF + ShH, high-fat diet mixed with 6% *S. horneri*. Each value represents the mean ± SEM (*n* = 6−8). Significant difference: * *p* < 0.05, ** *p* < 0.01 vs. HF group, ## *p* < 0.01 vs. Normal group.

**Figure 5 nutrients-13-00551-f005:**
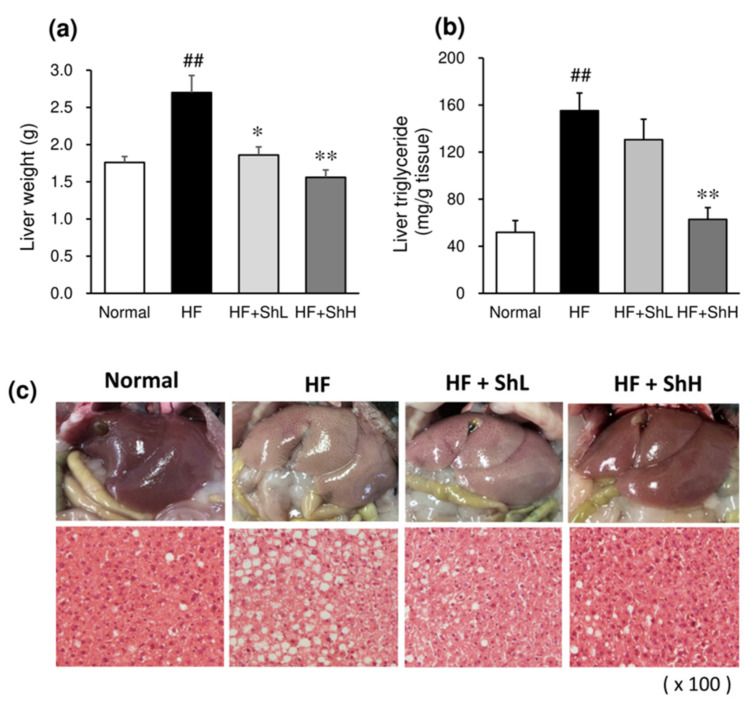
Effects of *S. horneri* on the hepatic lipid accumulation in C57BL/6J mice fed a high-fat diet for 13 weeks. (**a**) Liver weight; (**b**) liver triglyceride content; (**c**) representative gross morphology, and histological sections of the liver. Normal, normal diet; HF, high-fat diet; HF + ShL, high-fat diet mixed with 2% *S horneri*; HF + ShH, high-fat diet mixed with 6% *S. horneri*. Each value represents the mean ± SEM (*n* = 8). Significant difference: * *p* < 0.05, ** *p* < 0.01 vs. HF group, ## *p* < 0.01 vs. Normal group.

**Figure 6 nutrients-13-00551-f006:**
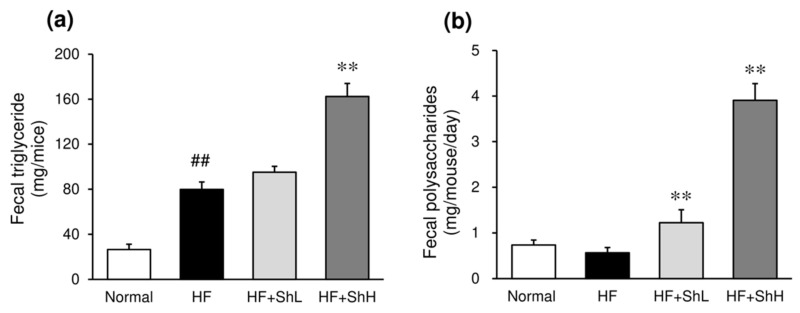
Effects of *S. horneri* on the feces components in C57BL/6J mice fed a high-fat diet. Feces were collected from each mouse 10 weeks after high-fat diet ingestion, and the contents of triglyceride (**a**), and polysaccharide (**b**) were measured after extraction with isopropanol and distilled water, respectively. Normal, normal diet; HF, high-fat diet; HF + ShL, high-fat diet mixed with 2% *S. horneri*; HF + ShH, high-fat diet mixed with 6% *S. horneri*. Each value represents the mean ± SEM (*n* = 6−8). Significant difference: ** *p* < 0.01 vs. HF group, ## *p* < 0.01 vs. Normal group.

**Figure 7 nutrients-13-00551-f007:**
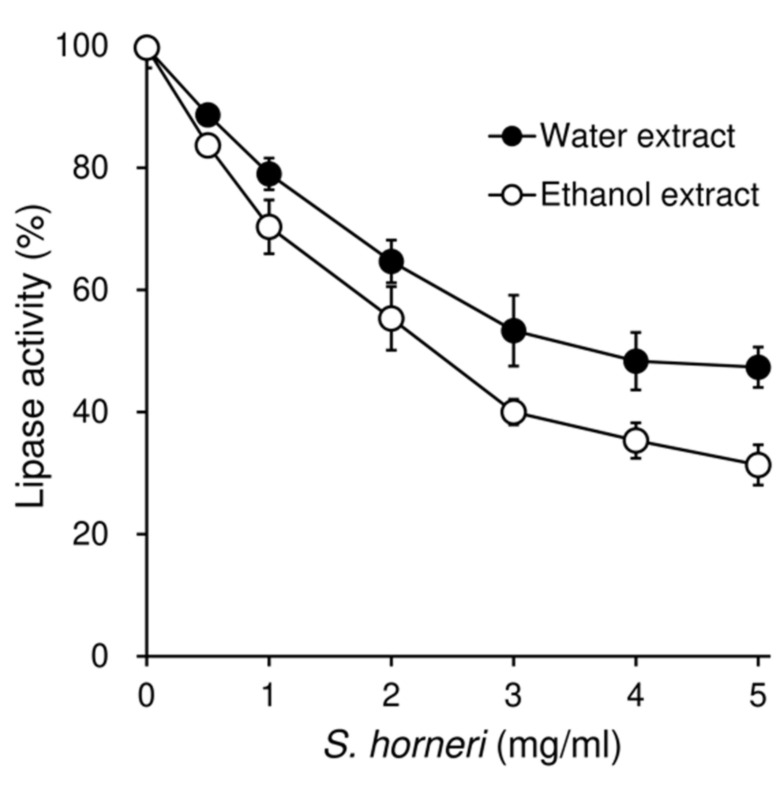
Effects of *S. horneri* extracts on the activity of pancreatic lipase in vitro. *S. horneri* was extracted with water (water extract) or 70% ethanol (ethanol extract), and its inhibitory activity was measured in vitro. Each point represents the mean ± SEM of triplicate experiments.

**Table 1 nutrients-13-00551-t001:** Composition of the experimental diets.

Ingredients (g/100 g)	Normal Diet	High-Fat (HF)	HF + ShL (2%)	HF + ShH (6%)
Cornstarch	46.57			
α-Cornstarch	15.5	16.0	15.0	13.0
Maltodextrin		6.0	6.0	6.0
Sucrose	10.0	5.5	5.5	5.5
Casein	14.0	25.6	25.6	25.6
Cellulose	5.0	6.6	5.6	3.6
Soybean oil	4.0	2.0	2.0	2.0
Lard		33.0	33.0	33.0
Mineral mix	3.5	3.5	3.5	3.5
Vitamin mix	1.0	1.0	1.0	1.0
L-Cysteine	0.18	0.36	0.36	0.36
Cholin bitartrate	0.25	0.25	0.25	0.25
Calcium carbonate		0.18	0.18	0.18
*S. horneri* powder			2.0	6.0

**Table 2 nutrients-13-00551-t002:** Effects of *S. horneri* on serum lipid levels.

	Normal	HF	HF + ShL	HF + ShH
Total cholesterol (mg/dL)	150 ± 9	217 ± 6 ^##^	215 ± 7	189 ± 7 *
HDL cholesterol (mg/dL)	86 ± 2	88 ± 2	85 ± 2	87 ± 2
Non-HDL cholesterol (mg/dL)	64 ± 7	128 ± 7 ^##^	124 ± 6	99 ± 5 **
Triglyceride (mg/dL)	69 ± 8	48 ± 6	33 ± 3	45 ± 4

Each value represents the mean ± SEM for 7–8 mice. Normal, normal diet; HF, high-fat diet; HF + ShL, high-fat diet mixed with 2% *S horneri*; HF + ShH, high-fat diet mixed with 6% *S. horneri*. Significantly different from the HF group, * *p* < 0.05; ** *p* < 0.01. Significantly different from the Normal group, ## *p* < 0.01.

**Table 3 nutrients-13-00551-t003:** Effects of *S. horneri* on the serum parameters of the liver function.

	Normal	HF	HF + ShL	HF + ShH
ALT (IU/L)	32.7 ± 6.7	126.7 ± 23.5 ^##^	102.4 ± 17.2	40.1 ± 4.9 **
AST (IU/L)	107.4 ± 10.6	155.0 ± 17.4 ^##^	120.3 ± 10.6	103.0 ± 7.8 *
ALP (IU/L)	223.7 ± 12.3	287.3 ± 27.1 ^##^	255.8 ± 18.6	195.6 ± 8.8 **
LAP (IU/L)	33.8 ± 1.4	43.6 ± 3.7 ^#^	40.6 ± 2.7	32.1 ± 0.93 *

Each value represents the mean ± SEM for 7–8 mice. Normal, normal diet; HF, high-fat diet; HF + ShL, high-fat diet mixed with 2% *S horneri*; HF + ShH, high-fat diet mixed with 6% *S. horneri*. ALT, alanine aminotransferase; AST, aspartate aminotransferase; ALP, alkaline phosphatase; LAP, leucine aminopeptidase. Significantly different from the HF group, * *p* < 0.05; ** *p* < 0.01. Significantly different from the Normal group, # *p* < 0.05; ## *p* < 0.01.

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
