# Peer review of "The Edible Brown Seaweed Sargassum horneri (Turner) C. Agardh Ameliorates High-Fat Diet-Induced Obesity, Diabetes, and Hepatic Steatosis in Mice"

_nutrients, 2021, doi:10.3390/nu13020551_

Round 1

Reviewer 1 Report

General comment: The authors presented an interesting original work concerning to the role of Sargassum horneri (Turner) C. Agardh (S. horneri) in the obesity, diabetes and hepatic steatosis induced by the intake of a high-fat diet. For this, the researchers used mice as model of disease.

The manuscript should be revised by an English native.

Title: It is short, clear and concise.

Abstract: It is adequate.

Introduction: It is adequate.

Materials and Methods:

The methods should be described chronologically. E.g. The preparation of the powder and extracts of S. horneri should be described before the

Please indicate the exact number of animals per group.

Were the animals allocated in the same cage or individually?

Were the animals supplemented with the S. horneri for 13 weeks? Please clarify.

Were the blood samples obtained from each vein? Or were they obtained directly from the heart? Please clarify. How were the samples stored?

“Liver tissue and epididymal, peritoneal, and mesenteric white adipose tissues were removed, weighed, and stored at -80 °C.” How were the samples obtained? Were the animals sacrificed?

“Some of the mice were used for histological examinations.” How many animals per group? Please clarify.

Results: The figure 1b was not full visible.

In a general way, the Results are properly presented.

Discussion: It is adequate.

References: They are adequate.

Recommendation: The manuscript should be accepted for publication after a Moderate revision.

Author Response

Materials and Methods

As the last part of the sentence is missing, we cannot understand this comment. But we judged that "Preparation of powder and extracts of S. horneri" should be placed before the "Animal experiments and dietary treatment". So, we have moved "Preparation of powder and extracts of S. horneri" forward (lines 73-81, written in red).

The manuscript has been proofread by a native speaker of English.

The exact number of animals has been added (lines 86-88).

Four animals were allocated in each cage.

The periods of supplementation has been described (line 94; After 13 weeks of --).

The blood samples were withdrawn from the ophthalmic vein. The way of blood collection has been added (line 95).

As described in lines 86-89, mice were anesthetized and the blood was collected from the ophthalmic vein. Then, mice were sacrificed and the tissues were removed.

We have added the number of mice (n=3-4) for histological examinations (line 89).

Results

We have adjusted the position of figure 1.

Reviewer 2 Report

The aim of the current study was to assess the impact of S. horneri on obesity, diabetes, hepatic steatosis, and hypercholesterolemia in mice who were put high-fat diet. Authors found that supplementation of mice with S. horneri suppressed high-fat diet-induced body weight gain and the accumulation of fat in adipose tissue and liver, and the elevation of the serum glucose level.

Major comments

The aim of the study is to consider with a relevant clinical implication, the design is valid, as well as the presentation of results is clear. However my major comment is that it seems that authors forget that this study has been conducted in mice, and they jump to the conclusions that the supplementation of S. horneri is a beneficial foodstuff with potential therapeutic applications (as mentioned in the abstract in lines 34-35).

I completely disagree with this affirmation. I strongly recommend authors especially in the abstract, introduction and discussion as well as conclusion section, to mention that this study has been conducted in mice, and the translation of these findings in human should be demonstrated. Moreover authors should be aware also that only 10% of animal studies are translated in humans. Otherwise this study will create a huge false expectation for both clinicians as well as patients.

Therefore this raised issue should be clearly and extensively addressed with suitable references in the sections as follows:

  • Abstract: at least 1-2 statements
  • Introduction: At least 1 paragraph
  • Discussion: At least a couple of paragraphs
  • Conclusions: At least 2-3 statements

I expect substantial changes in this direction. 

Author Response

We understand that this study was performed in the experimental animal. Some expressions concerning the anticipated effect of S. horneri in humans may not be appropriate. We have revised the manuscript according to the reviewer's comments:

We have deleted or revised the following sentences concerning the effectiveness of S. horneri in humans.

Abstract: "Thus, S. horneri is a beneficial foodstuff with potential therapeutic applications." has been deleted (lines 29-30).

Introduction: "several studies have ---" has been revised to "several studies using animal models have ---" (line 54).

Reviewer 3 Report

In the present manuscript the authors have evaluated the effect of S.horneri extract in the amelioration of obesity and related diseases. They have shown that if HF diet is supplemented with S.horneri extract and fed to mice for 13 weeks that results in decreased body weight. They have also done several other experiments to prove the effect of S.horneri extract to ameliorate obesity.

Overall, I think the article is well-written, the experiments have been thorough and the results have been well-communicated. Other than minor  formatting errors (Lines 125, 230).

Author Response

The suggested point (line 230) has been revised (line 229) in revised manuscript, written in red).

We cannot find formatting error in line 125.

Round 2

Reviewer 1 Report

The manuscript should be accepted for publication in the present form.

Author Response

There is no new comment about the revision.

Reviewer 2 Report

Again authors should be responsive to all my comments. You cannot ignore the comments, please adhere to my previous comments.  

Author Response

Based on the reviewer's comments that this study was carried out in mice, we have revised the content of our manuscript again. Furthermore, we have added some sentences in order not to mix up animal experiments with human trials.

Abstract (previous revision; the last part of the Abstract): "Thus, S. horneri is a beneficial foodstuff with potential therapeutic applications." has been deleted.

Introduction

1.  Most of papers concerning biological and pharmacological activities of various compounds were performed in experimental animals, not in humans. The cited references [10-13] are about experimental data performed in animal models. So, we have added "in experimental animal models" (line 48) to the last part of the sentence. 

2. We have added "human data are ---" (lines 49-50).

3. The references 16 and 17 are data performed in experimental animals. We have added "using animal models" (line 55).

4. We have added "in animal and in vitro studies"", because the references 18 and 19 are about experimental data performed in animal models and in vitro (line 60).

5. We have newly added the results in humans as references 23 and 24 (lines 63-65).

Discussion

1. We have added "This animal experiment revealed that" (line 290). 

2. We have added "In vitro and animal studies have suggested that" (lines 295-296).

3. We have newly added the results obtained in human trials "A randomized, double-blind, ---" (lines 297-299).

4. We have added "in HF-induced obese mice" (line 307).

5. We have revised to "The present in vitro study" (line 339). 

6. We have added the results obtained in human trials, "Sixteen-week, double blind, ---"  (lines 372-373).

Conclusions

We have deleted "Thus, the traditionally eaten brown seaweed S. horneri may be useful as a foodstuff for reducing rate of obesity and diabetes", and have newly added the following sentence to emphasize that this study was performed in mice;"This study involved experiments using mice, so the results cannot be directly applied to humans. Further studies are needed to demonstrate the effectiveness of S. horneri in humans.".

Due to above revisions, we have added 4 new references (Ref. 23, 24, 31, and 53).